# Defining dual-axis landscape gradients of human influence for studying ecological processes

**Benjamin Juan Padilla**[1], **Chris Sutherland**[2]*

**1** Research Institute – Indiana University of Pennsylvania, Indiana, Pennsylvania, United States of America,
**2** Centre for Research into Ecological and Environmental Modelling, University of St Andrews, St Andrews,
United Kingdom

* css6@st-andrews.ac.uk

**Data Availability Statement:** Data and code to reproduce analysis are available on OSF with the following citation. Padilla BJ, Sutherland C. Defining dual-axis landscape gradients of human influence for studying ecological processes

## Abstract

Ecological processes are strongly shaped by human landscape modification, and understanding the reciprocal relationship between ecosystems and modified landscapes is critical for informed conservation. Single axis measures of spatial heterogeneity proliferate in the contemporary gradient ecology literature, though they are unlikely to capture the complexity of ecological responses. Here, we develop a standardized approach for defining multi-dimensional gradients of human influence in heterogeneous landscapes and demonstrate this approach to analyze landscape characteristics of ten ecologically distinct US cities. Using occupancy data of a common human-adaptive songbird collected in each of the cities, we then use our dual-axis gradients to evaluate the utility of our approach. Spatial analysis of landscapes surrounding ten US cities revealed two important axes of variation that are intuitively consistent with the characteristics of multi-use landscapes, but are often confounded in single axis gradients. These were, a hard-to-soft gradient, representing transition from developed areas to non-structural soft areas; and brown-to-green, differentiating between two dominant types of soft landscapes: agriculture (brown) and natural areas (green). Analysis of American robin occurrence data demonstrated that occupancy responds to both hard-to-soft (decreasing with development intensity) and brown-to-green gradient (increasing with more natural area). Overall, our results reveal striking consistency in the dominant sources of variation across ten geographically distinct cities and suggests that our approach advances how we relate variation in ecological responses to human influence. Our case study demonstrates this: robins show a remarkably consistent response to a gradient differentiating agricultural and natural areas, but city-specific responses to the more traditional gradient of development intensity, which would be overlooked with a single gradient approach. Managing ecological communities in human dominated landscapes is extremely challenging due to a lack of standardized approaches and a general understanding of how socio-ecological systems function, and our approach offers promising solutions.

[Internet]. OSF; 10 Aug 2021. Available: osf.io/
29ct4.

**Funding:** The author(s) received no specific
funding for this work.

**Competing interests:** The authors have declared
that no competing interests exist.

## Introduction

Rapid expansion of the global human population has led to increasing concern for natural systems and biodiversity. Anthropogenic landscape modification profoundly influences resource availability and habitat quality, which in turn, determines patterns of species distribution and abundance [1]. Given the explicit link between patterns of landscape structure and ecological processes, and the extent of human modification to the landscape, informed conservation and ecosystem management requires reliable descriptors of landscape heterogeneity gradients with an anthropogenic focus [2, 3]. Nevertheless, well documented variability in the quality, complexity, and ecological relevance of quantitative measurements of landscape structure have contributed to a lack of a general and scalable understanding of how ecological processes respond to landscape heterogeneity, particularly along gradients of human modification [3–6].

The need for ecologically meaningful measures of landscape heterogeneity (i.e., composition and configuration of landscape features) to understand drivers of ecosystem responses is well recognized [7], and over time numerous conceptual, theoretical, and applied approaches have been posited [4, 5, 8, 9]. These approaches range from the patch mosaic (fragmentation) paradigm, which, while valuable in some contexts, is arguably overly simple in heterogenous landscapes [10–12], to various metrics of patch complexity and distribution [13]. Efforts to improve the ecological relevance and realism of landscape metrics has led to the development of models thought to better represent the continuous nature of landscape heterogeneity and ecological processes by extending the patch-centered perspective to incorporate the composition of the surrounding landscape [14]. Regardless of the metrics used, successful integration of ideas in spatial ecology across systems and scales requires an improved appreciation for what landscape descriptors are measuring, and how they relate to ecological processes [15]. That is, reliable and accurate measures of landscape heterogeneity are a prerequisite to understanding patterns of ecological response across scales.

Efforts to understand and quantify ecological responses across anthropogenic gradients has resulted in some general, though equivocal, predictions about patterns of ecological response to spatial heterogeneity in human dominated landscapes. For instance, a negative relationship between species richness and human disturbance has been demonstrated in birds [16, 17], invertebrates [18, 19], plants [20, 21], and other taxa [22, 23]. Moreover, this relationship is often non-linear, with a peak in richness in areas of intermediate human modification [24, 25]. At the species level, however, responses vary, and depend on the ecology of the species in question [26–28]. While fragmentation and human population density have been linked to decreases in movement and home range size in many species [29–32], much of the literature suggests no relationship [33–35], or uncertain relationships [36, 37] between a range of ecological processes (e.g., population size, species distribution) and landscape change. These apparent contradictions suggest that measured responses to gradients of landscape heterogeneity are context or locale specific and has led to calls for improved measures of human-dominated landscapes that move towards a more general understanding of ecological dynamics in human-dominated ecosystems [6, 38].

Attempts to improve the applicability and scalability of landscape metrics used in ecological analyses has led to almost exclusive use of one-dimensional gradients of variation (e.g., percent impervious surface), even though landscape heterogeneity, and in particular the myriad ways humans alter landscapes, is multi-dimensional [39, 40]. Highly dimensional landscapes, when compressed into one-dimensional descriptors, are likely to fall short in terms of ecological realism, i.e., the landscape as perceived by a species or community, limiting the ability to infer links between landscape patterns and ecological processes, with important consequences regarding how ecosystem processes are understood and managed in the Anthropocene. We

propose an extension of the typical one-dimensional approach, which involves the identification of multiple axes of landscape heterogeneity in the context of human influence.

In this paper, we develop a multi-dimensional approach to defining landscape heterogeneity that can be used for making inferences about species distributions in human dominated landscapes. We demonstrate the generality of our multi-dimensional gradient approach by jointly analyzing urban-exurban landscapes in ten geographically and ecologically distinct US cities, identifying two significant and biologically relevant axes of variation. We demonstrate the utility of our approach in a case study analysis of American robin (*Turdus migratorious*) occupancy. Specifically, by jointly analyzing detection—non-detection data from the same ten landscapes, we investigate continental-scale consistencies in species responses to two gradients of human influence.

## Methods

We selected ten geographically distinct medium sized cities (population between 200,000 and 500,000), widely distributed throughout the contiguous United States, representing the Level I ecoregions as defined by the U.S. Environmental Protection Agency [41]. These were Worcester (Massachusetts), Lexington (Kentucky), Jackson (Mississippi), Lincoln (Nebraska), Lubbock (Texas), Salt Lake City (Utah), Albuquerque (New Mexico), Bakersfield (California), Portland (Oregon), and Spokane (Washington, Fig 1). For each city, we extracted 30-m resolution landcover data from the freely available 2016 National Land Cover Database [42] for a 50-by-50 kilometer window surrounding the city center (coordinates extracted from www.latlong.net). This spatial extent extends well into exurban regions and thus represented the full extent of landscape heterogeneity for each city. To test for sensitivity to the extent, we repeated the analysis at alternative windows and found no difference in our results (*Effects spatial extent*: *30 x 30 km city window* in S1 File).

Landscape composition was fairly evenly split between three dominant lands cover categories when aggregated among all cities: developed (20.03%), forests (23.98%), and agriculture (31.28%) and contained fifteen of the nineteen Anderson Land-Cover classes used by the NLCD. The remaining four ('Perennial Ice-Snow', 'Dwarf Scrub', 'Sedge-Herbaceous',

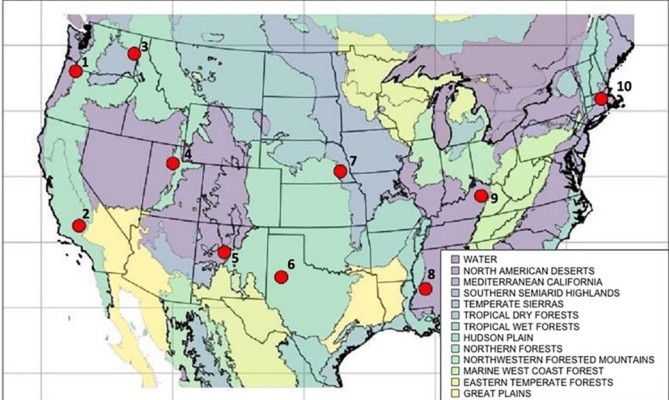

**Fig 1. Map of 10 study cities.** Map showing the locations of all study cities for the landscape quantification and ecological case study. Background colors represent unique Level 1 EPA Eco-Regions. Study cities are represented by numbered red points. 1—Portland, OR, 2—Bakersfield, CA, 3—Spokane, WA, 4—Salt Lake City, UT, 5—Albuquerque, NM, 6—Lubbock, TX, 7—Lincoln, NE, 8—Jackson, MS, 9—Lexington, KY, and 10—Worcester, MA. Ecoregion GIS data was sourced from the US EPA—Ecoregion spatial database (https://www.epa.gov/eco-research/ecoregions-north-america). Map was produced using the package 'map' in R.

'Lichen') are restricted to Alaska or high elevation locations. At the individual city level, landscape composition was more variable; forested classes dominated Worcester and Spokane (39.08%, 30.22%), Albuquerque was largely scrubland (46.81%), Lexington dominated by pasture (62.31%), and agriculture in Lincoln, Bakersfield, and Lubbock (47.61%, 42.2%, 72.15%). Details for each city are provided in Table 1.

Landscape analyses followed the landscape quantification framework outlined by Padilla and Sutherland [3]. Our decisions regarding the types of landscape features relevant for analysis, the data to represent those features, and the spatial scales of analysis were made to reflect a typical ecological analysis—definitions of, and justification for, these decisions are provided in Table 2. In general, the landscapes within which our cities were set were characterized by a mosaic of natural (forests and wetlands) and un-natural (crop and developed) land-cover categories which are captured well in the NLCD classification system.

The NLCD is a 30-m resolution raster dataset where each landscape pixel is classified as a single cover type. Ecosystems are influenced both by characteristics of a fixed location, and by

**Table 1. Table of study cities.**

| City, State | Population | Level I Ecoregion | Open Water | Devel. | Forests | Scrub Grass | Crop Pasture |
|---|---|---|---|---|---|---|---|
| **Worcester, MA** | 185,877 | ER5 –Northern Forests | 3.33 | 23.15 | 65.51 | 2.54 | 6.11 |
| **Spokane, WA** | 208,916 | ER6 –NW Forested Mountains | 1.23 | 14.46 | 31.31 | 31.91 | 20.62 |
| **Salt Lake City, UT** | 200,591 | ER6 –NW Forested Mountains | 11.02 | 23.00 | 36.37 | 23.34 | 5.29 |
| **Portland, OR** | 583,776 | ER7 –Marine West Coast Forest | 3.12 | 37.06 | 23.43 | 7.85 | 28.54 |
| **Lexington, KY** | 323,780 | ER8 –Eastern Temperate Forests | 0.56 | 15.31 | 16.65 | 1.03 | 66.44 |
| **Jackson, MS** | 164,422 | ER8 –Eastern Temperate Forests | 4.85 | 30.11 | 43.29 | 12.16 | 18.87 |
| **Lubbock, TX** | 255,885 | ER9 –Great Plains | 0.15 | 12.58 | 0.23 | 14.89 | 72.15 |
| **Lincoln, NE** | 287,401 | ER9 –Great Plains | 1.54 | 13.02 | 5.86 | 29.4 | 50.82 |
| **Albuquerque, NM** | 560,218 | ER10 –North American Deserts | 0.23 | 17.66 | 15.96 | 67.71 | 3.37 |
| **Bakersfield, CA** | 383,679 | ER11 –Mediterranean | 0.51 | 13.96 | 1.19 | 37.93 | 46.41 |

List of ten urban-exurban regions used for landscape comparisons, including population size (2010 census) and US-EPA Ecoregion. Values for land cover types represent the percent coverage in a given city.

**Table 2. Landscape analysis decision table.**

| | Decision | Justification |
|---|---|---|
| 1) Landscape Features | Physical land-cover and demographic land-use | 'Land-cover' categories (i.e. forest, shrub) track changes in 'natural' landscapes, while 'land-use' (devel., crop) tracks the human footprint and approximate population density |
| 2) Spatial Data | Remote-sensed, National Land Cover Data (2016) | NLCD land-cover data is readily available and is a consistent data-source to represent landscape features in all 10 study cities |
| 3) Spatial Scale | 500-m and 1,500-m Gaussian kernel | Spatial extent (50 x 50-km) chosen to capture sufficient spatial and ecological heterogeneity. Primary spatial grain (500-m kernel) selected to represent breeding home range of American robin. 1,500-m as a common scale in ecological research selected to compare effects of scale. |

Decisions made within landscape gradient framework for analyzing urban landscapes in jointly across study cities and in the city-specific analysis. This follows the framework outlined in Padilla & Sutherland 2019. Justification provided here is in light of dual research goals. First, to quantify landscape pattern in 10 distinct cities, and second, to evaluate occupancy patterns of American robin in response to landscape gradients.

the local landscape context surrounding a given location [43]. Therefore, for each NLCD land-cover class, we extracted a binary raster surface (1 if focal class, 0 if otherwise), and to account for landscape surrounding a given location (i.e., landscape context) we computed the spatially weighted average for each pixel using a Gaussian kernel spatial smooth, resulting in a continuous surface ranging from 0 (no focal class within smoothing kernel) to 1 (smoothing neighborhood entirely focal class). The width of the kernel defines the spatial grain of analysis, and therefore should be selected with the research specific ecological process in mind [44]. We selected a 500-m smoothing kernel for our analysis based on the typical breeding home range size of our case study focal species, the American robin [45]. We tested sensitivity of landscape quantification to this choice by replicating the analysis with a 1500-m spatial scale and found no effect of scale selection of downstream inference (*Effects of smoothing scale* in S1 File). All processing of the spatial data was conducted in R Version 3.5.3 [46] using the 'raster' [47], 'FedData'[48] and 'smoothie' [49] packages.

To identify dominant patterns of variation in these landscapes, we used Principal Components Analysis (PCA). PCA is one of several methods for summarizing a large number of potentially correlated variables into fewer uncorrelated axes of variation (others include factor analysis, non-metric multidimensional scaling, correspondence analysis), and it is particularly well suited to exploratory ordination and gradient analysis [50]. Using a matrix of class-specific smoothed landscape variables, we conducted PCA on the data for all cities combined. Dominant principal components were identified and selected based on a cumulative weight cut-off of the broken stick method, which retains components that explain more variance than would expected than dividing variance randomly among all components [51]. These were used to produce a spatially explicit gradient of habitat heterogeneity based on the resolution of the input data, where the value for each pixel in the resulting raster surface is the PCA weighted average calculated as the sum of that pixel's smoothed NLCD values multiplied by the corresponding PC weight for each NLCD value. We also conducted this analysis for each city independently in order to determine how well the combined (i.e., all cities) gradients described city-specific gradients. Output for our PCA analyses are reported in the Results section under *Landscape Gradient Analysis*.

## Ecological case study

We evaluated the utility of multi-dimensional landscape heterogeneity gradients for ecological applications using a real-world case study. Specifically, using occupancy modelling we tested whether simultaneous consideration of multiple landscape gradients alters inferences about ecological responses relative to the traditional single-gradient approach. We analyzed American robin detection-nondetection data under an occupancy modelling framework using the gradients as covariates. We selected the American robin because it is a widespread generalist species, present in all ten focal cities, and because it is widely considered to be human-adaptive.

Robin detection histories were analyzed using a single-season hierarchical occupancy model, which estimates site occupancy probabilities while accounting for imperfect detection [52]. Stationary, complete checklists in which non-reporting of a species assumed to be non-detection, from surveys conducted from April 1st through September 30th 2018 were extracted from the *eBird* online database [53] using the R package 'auk' [54]. In this analysis, detection data from all cities were pooled in a single analysis. Because there was substantial variation in the number of eBird locations in each city (i.e., each unique fixed eBird survey site), and to improve balance and reduce regional bias in sample size, the data were randomly thinned to a maximum of 250 locations (Table 1 in S2 File).

The standard occupancy model consists of two sub-models: a logit-linear model describing site- and occasion-specific detection probability ($p$), which can be modelled using site- and occasion-specific covariates, and a second logit-linear model describing site-specific occupancy probability ($\psi$), that can be modelled using site-specific covariates. To account for variation in detection, we considered the following covariates: city (categorical factor), sampling date, and date$^2$ to allow for peaks or troughs in detection, and site-specific landscape gradient values. Sampling date was scaled (0–18) such that a one unit increase in the date variable represented 10 calendar days, which facilitated parameter interpretation and model convergence. A total of 26 possible detection models were considered, which included all additive combinations and only single interaction terms (Table 2 in S2 File). For occupancy, we included the effect of city, again as a factor, each of the site-specific landscape gradient values, and all combinations of city-gradient interactions for a total of 16 candidate models (Table 2 in S2 File).

We adopted a two-stage modeling approach whereby we fit and compared all possible combinations of detection covariates, each with the 'global' (most complex) model for the occupancy component [55]. Using Akaike's Information Criterion (AIC) to rank models, the best supported model for detection was carried over to the second stage, where we compared competing models for occupancy. Finally, the model selected for inference was validated by examining model residuals and performing goodness of fit tests. Occupancy analysis was conducted in the package 'unmarked' [56], while AIC model selection and goodness of fit tests were done using the 'AICcmodavg' package [57]. All analyses, were conducted in R Version 3.5.3 [46].

## Results

### Landscape gradient analysis

Principal components analysis of the combined (i.e., all cities) landscape data identified three axes of variation, explaining 37.1% of the cumulative variance in the data (Table 3). When each city was considered independently, the same three axes explained between 42.60 and 54.89% of the variance (Table 1 in S1 File), demonstrating the scalability of emergent landscape gradients across scales. However, using the broken stick method [51], only the first two axes exceeded the 22.1% cumulative variance threshold for combined and city-specific analyses. The principal component explaining the largest proportion of data variation for the combined data (16.7%) was strongly negative for developed land-cover classes, with neutral or positive loadings for forested, open, and agricultural classes (Table 3). Developed classes are characterized by a high degree of impervious surface, buildings, and associated human population density, whereas the others are predominantly non-impervious natural (wetlands) or un-natural (pasture) landscapes. Thus, this first descriptor of landscape pattern can be interpreted as a transition from *hard* (characterized by impervious and human presence) to *soft* (unpaved natural or agricultural), which we refer to as a *hard-to-soft* gradient.

The second principal component explained 11.1% of the variation and showed a strong differentiation between the land use classes at the soft end of the hard-to-soft gradient. Specifically, this axis distinguishes between human modified but un-developed areas (cultivated croplands) and more natural areas (forests or wetlands). This axis is intuitively interpretable as a shift from modified agricultural landscapes, to un-developed natural regions, or, *brown*-to-*green*. While the hard-to-soft axis does not distinguish between dominant types of *soft* landscapes, the second accounts for this variation between *brown* and *green* regions and is a valuable counterpart to component one producing a triangular distribution (Fig 2).

The third principal component explained 9.3% of the total variation and was not retained to produce a gradient surface as the cumulative variance of the first two principal components exceeded the broken stick cutoff. However, it is interesting in that like PC2, the third principal

**Table 3. Dominant principal component axes.**

| NLCD Layer | | Obs. Freq. | PC1 | PC2 | PC3 |
|---|---|---|---|---|---|
| | | *Std.Dev.* | 1.581 | 1.295 | 1.182 |
| | | *Variance Explained (%)* | 16.7 | 11.1 | 9.3 |
| *Water* | 11—Open Water | 2.70 | 0.042 | 0.030 | -0.04 |
| *Developed* | 21—Developed Open | 6.49 | **-0.360** | 0.055 | 0.017 |
| | 22—Developed Low Intensity | 6.72 | **-0.545** | 0.047 | -0.040 |
| | 23—Developed Medium Intensity | 4.78 | **-0.553** | 0.015 | -0.125 |
| | 24—Developed High Intensity | 1.61 | **-0.392** | -0.007 | -0.148 |
| *Barren* | 31—Barren Land | 0.79 | 0.039 | -0.017 | -0.116 |
| *Forest* | 41—Forest Deciduous | 10.17 | 0.119 | **0.469** | 0.171 |
| | 42—Forest Evergreen | 8.96 | 0.154 | 0.269 | **-0.349** |
| | 43—Forest Mixed | 2.14 | 0.087 | **0.433** | 0.001 |
| *Shrubland* | 52—Scrub/Shrub | 10.56 | 0.151 | -0.012 | **-0.554** |
| *Herbaceous* | 71—Grassland/Herbaceous | 10.85 | 0.140 | **-0.339** | -0.282 |
| *Cultivated* | 81—Pasture/Hay | 11.25 | 0.053 | 0.082 | **0.446** |
| | 82—Cultivated Crops | 19.41 | 0.119 | **-0.491** | **0.387** |
| *Wetlands* | 90—Woody Wetlands | 2.81 | 0.043 | **0.378** | 0.157 |
| | 95—Herbaceous Wetlands | 0.77 | 0.032 | 0.075 | -0.039 |

Dominant Principal Component axes. The Obs. Freq. column displays the percent composition of each land cover category in all cities combined. For each principal component, standard deviation, percent variance explained, and rotated variable loadings are displayed. Variables with a strong weight are in bold. The first two axes were selected because cumulative variance exceeded the 22.1% broken stick threshold.

component reflected a divergence between modified and un-modified undeveloped areas. While PC2 differentiated natural deciduous and mixed forests from modified croplands, the third axis is a gradient from evergreen forests and scrub, to pastures (Table 3). Both PC2 and PC3, therefore, can be interpreted as brown-to-green in different habitat and land-use types.

Due to the ecological complementarity of the two dominant components, and our focus on highlighting the value of simultaneously considering multiple dimensions of human influence, our approach considers these axes jointly. However, it is worth noting that on their own, these gradients are analogous to traditional approaches that consider single gradients in isolation. The hard-to-soft gradient is consistent with traditional urban gradients focusing on the built environment (e.g., percent impervious surface or housing density) [3, 6, 58], or it's complement, percent forest cover. The more agricultural brown-to-green gradient, though less common in urban ecology, has been used in agro-ecological investigations [59, 60]. Our approach allows us to investigate ecological responses to both important characteristics of human influence simultaneously.

As a test of whether these axes were consistent locally and at varying spatial extents, we conducted the same analyses of NLCD data for each city independently as well as jointly using a 30x30 km window. Both city specific, and 30x30 km analyses revealed the same dominant axes of variation as the 50x50 km combined analysis. As expected, the component weights of NLCD classes and absolute values of axes differed, nevertheless, interpretation of these axes remained consistent (*Effects of smoothing scale: 1,500-m scale* in S1 File).

## Ecological case study

Our robin analysis included data from a total of 1,703 sampling locations (sites) across all cities (min: 31 in Bakersfield, max: 250 in Worcester, Albuquerque, Portland, and Salt Lake City).

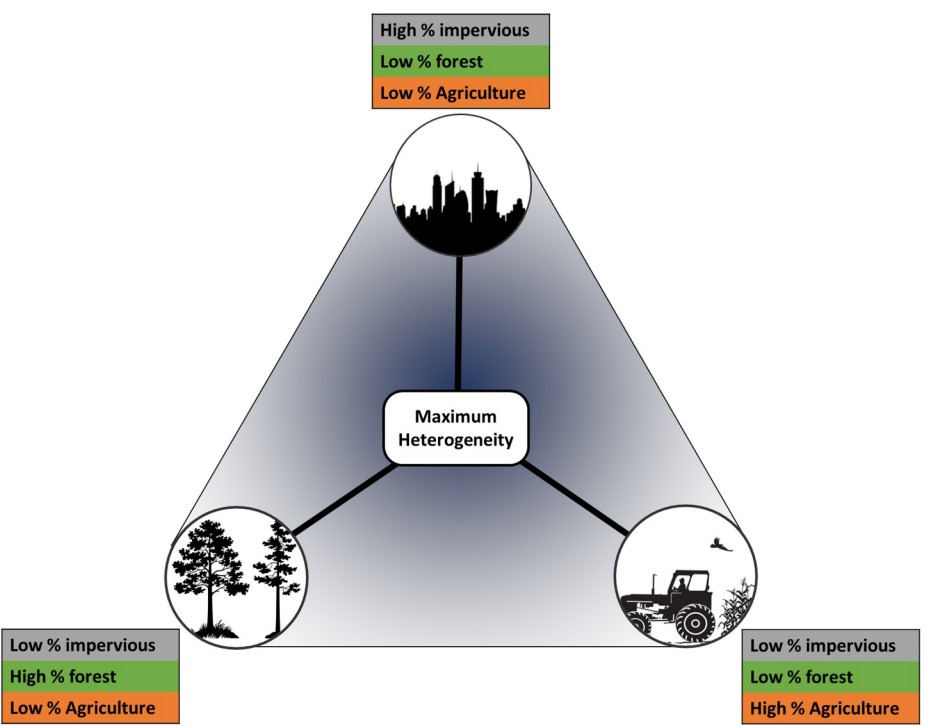

**Fig 2. Conceptual diagram of multi-dimensional landscape gradient: A conceptual description of the triangular distribution captured by a multi-dimensional landscape definition that differentiates between urban, agricultural, and natural portions of the landscape along dual axes of variation.** Hard and soft portions of the landscape are sorted along the vertical axis, while brown and green regions along the horizontal. This results in a multi-dimensional perspective where heterogeneity is maximized at the center of both axes.

There were a total of 5,779 sampling visits across all cities, with a mean number of visits per site of 1.95 (range: 1–172, Table 1 in S2 File). The overall proportion of sites with a minimum of one observation (i.e., naïve occupancy) was 0.43, which varied by city from 0.387 in Spokane, to 0.482 in Bakersfield (Table 1 in S2 File).

Of the 26 detection models considered, only eight converged, largely due to the complex model structure. The AIC-top model (AIC$_c$ wt = 1.0) included additive effects of both landscape gradients, a quadratic effect of date, and a city by date interaction term (Table 4). In the second step, we used the best supported detection model to evaluate 16 candidate occupancy models. Here, a single model held the majority of support (AIC$_c$wt = 0.91, Table 4) and included the effects of both landscape gradients, city, and an interaction between city and the hard-to-soft gradient. The second model was identical to the top model apart from the inclusion of one additional term, the interaction between city and brown-to-green. Given the lack of support for the additional terms, as indicated by the model ranking [61], model evaluation and inference that follows is based on the top model. Examination of model residuals and a Chi-Square goodness of fit test showed adequate model fit.

There was a significant quadratic effect (*estimate± SE)* of survey date on detectability (-0.012± 0.002), such that detection probability increased, reached a peak, and declined. Robin detection varied significantly along the *brown*-to-*green* axis, with robins more likely to be observed in more '*green'* landscapes (0.14± 0.05), and showed a negative relationship with hard-to-soft, though confidence intervals included zero (-0.38± 0.04). Date of peak detectability ranged from April 1$^{st}$ in Bakersfield (*date* = 0.0) to July 23$^{rd}$ in Portland (*date* = 11.3), while

**Table 4. Model selection table.**

| | Detection Model Structure | $K$ | $AIC_c$ | $\Delta AIC_c$ | $AICwt$ | - $LogLik$ |
|---|---|---|---|---|---|---|
| *1* | ~ city*date+date²+HS+BG <br> ~ ψ | 63 | 6544.02 | 0.0 | 1 | -3206.45 |
| *2* | ~ city*date+HS+BG <br> ~ ψ | 62 | 6611.83 | 67.81 | 0 | -3241.44 |
| *3* | ~ city*HS <br> ~ ψ | 60 | 6998.33 | 454.31 | 0 | -3436.84 |
| *4* | ~ city <br> ~ ψ | 43 | 7001.75 | 457.73 | 0 | -3456.69 |
| *5* | ~ date <br> ~ ψ | 42 | 7038.66 | 494.64 | 0 | -3476.20 |
| *6* | ~ BG <br> ~ ψ | 42 | 7450.53 | 906.51 | 0 | -3682.13 |
| *7* | ~ 1 ~ ψ | 41 | 7450.69 | 906.67 | 0 | -3683.27 |
| *8* | ~ HS <br> ~ ψ | 42 | 7452.32 | 908.30 | 0 | -3683.03 |
| | Occupancy Model Structure | $K$ | $AIC_c$ | $\Delta AIC_c$ | $AICwt$ | - $LogLik$ |
| *1* | ~ p    ~ city*HS+BG | 44 | 6530.44 | 0 | 0.91 | -3219.98 |
| *2* | ~ p    ~ city*(HS+BG) | 53 | 6535.21 | 4.77 | 0.09 | -3212.80 |
| *3* | ~ p    ~ city*HS | 43 | 6542.84 | 12.40 | 0 | -3227.23 |
| *4* | ~ p    ~ city*(HS*BG) | 63 | 6544.02 | 13.58 | 0 | -3206.45 |
| *5* | ~ p    ~ city+HS*BG | 36 | 6567.86 | 37.42 | 0 | -3247.10 |
| *6* | ~ p    ~ city+HS+BG | 35 | 6572.90 | 42.46 | 0 | -3250.67 |
| *7* | ~ p    ~ city+HS | 34 | 6578.90 | 48.46 | 0 | -3254.71 |
| *8* | ~ p    ~ city*BG+HS | 44 | 6580.24 | 49.80 | 0 | -3244.88 |
| *9* | ~ p    ~ HS*BG | 27 | 6580.55 | 50.11 | 0 | -3262.81 |
| *10* | ~ p    ~ city*BG | 43 | 6581.49 | 51.05 | 0 | -3246.56 |
| *11* | ~ p    ~ city+BG | 34 | 6582.03 | 51.59 | 0 | -3256.27 |
| *12* | ~ p    ~ city | 33 | 6587.86 | 57.42 | 0 | -3260.23 |
| *13* | ~ p    ~ BG | 25 | 6490.33 | 59.89 | 0 | -3269.76 |
| *14* | ~ p    ~ HS+BG | 26 | 6591.46 | 61.02 | 0 | -3269.29 |
| *15* | ~ p    ~ HS | 25 | 6600.95 | 70.51 | 0 | -3275.07 |
| *16* | ~ p    ~ 1 | 24 | 6601.16 | 70.72 | 0 | -3276.21 |

Model selection results for both detection and occupancy components of the American robin analysis based on sample size corrected AIC. K denotes the total number of parameters in the model and *AICwt* is the model weight. Detection was assessed with the global occupancy model and the best model for detection was used in all models for occupancy. Here, HS refers to the hard-to-soft gradient, while BG denotes brown-to-green.

maximum detection probability ranged from 0.33 in Worcester, to 0.87 in Jackson (Fig 3; Table 3 in S2 File).

Robin occupancy varied by city and with both gradients. Holding both gradients at 0 (overall scaled average), robin occupancy ranged from a low of 0.47 (0.08) in Albuquerque, New Mexico to a high of 0.99 (0.074) in Jackson, Mississippi. Robin occupancy was positively related to the brown-to-green axis (0.52± 0.01), suggesting that robins are more likely to occur in more forested areas than in areas characterized as predominantly open or agricultural. This effect was universal across all cities. In contrast, and interestingly, direction and magnitude of the hard-to-soft gradient effect varied by city, i.e., the responses to the gradient describing the transitions from built to vegetative environments was specific to each city (Fig 4). For example, occupancy was positively associated with the hard-to-soft gradient in Spokane (1.61± 0.819), but negatively associated with hard-to-soft in Worcester (-1.72± 0.765).

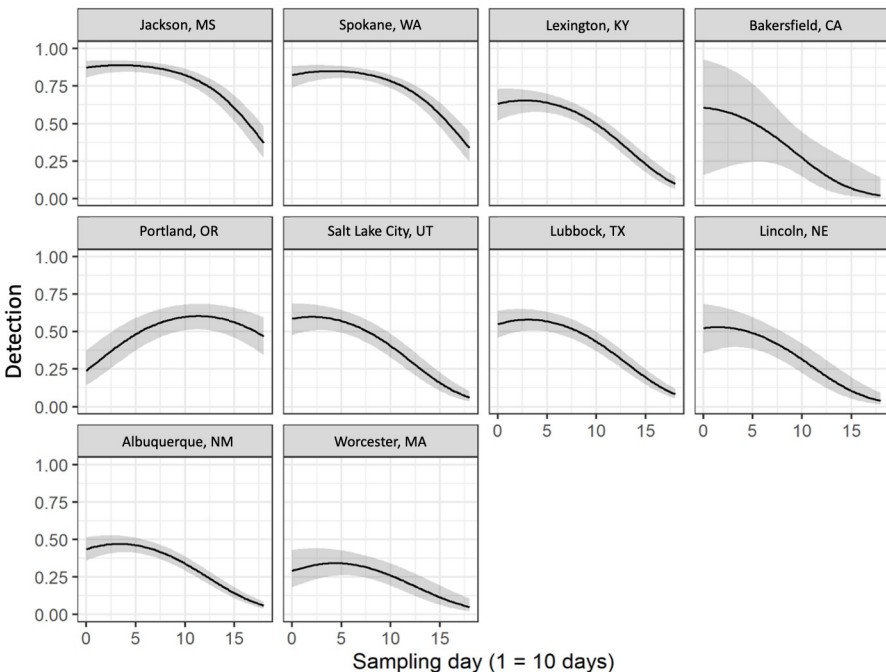

**Fig 3. Detection probability of American robin: Robin Detection probability as a function of survey date for each city predicted from the top model.** Grey shaded area represents 95% confidence intervals and solid is the expected value. Date of peak detection probability varied between cities, but tended toward the start of the study period, which coincides with robin breeding behavior.

## Discussion

Analysis of spatially heterogeneous landscapes surrounding ten metropolitan regions revealed two statistically important and ecologically intuitive axes of variation, which offers an exciting alternative to the conventional one-dimensional approach to investigating ecological responses in human-dominated landscapes. Despite regional variation in landscape composition (Table 1 in S1 File) the dual-gradient approach we present here consistently distinguished between two distinct types of anthropogenic influences: a *hard*-to-*soft* gradient capturing a continuum of the built human environment, and a *brown*-to-*green* gradient capturing the human agricultural footprint (Fig 2). Our analysis shows that in addition to being fundamental properties of the landscape, considering these axes jointly provides ecological insight that would otherwise be overlooked using a single-axis approach (Table 3). This multi-dimensional perspective highlights the importance of considering the complexity of human-dominated landscapes and identifies a triangular distribution of human influence that presents an intuitive and generalizable framework for understanding patterns of ecological function and developing management strategies in human-dominated landscapes.

Landscape metrics that are adaptable to a variety of ecosystem contexts are needed to improve understanding of human-dominated ecosystems and effectively synthesize local and regional conservation efforts. Prior attempts to produce universal metrics for human footprint or urbanization have thus far failed to result in methodological consistency or broad uptake, in part due to methodological complexity and data requirements. For example the HERCULES method [62] requires users to classify the landscape into categories of building, surface cover, and vegetation using LiDAR data. Seress et al. [63] describe another method that also requires some user based classification of satellite imagery into categories of buildings, vegetation, and

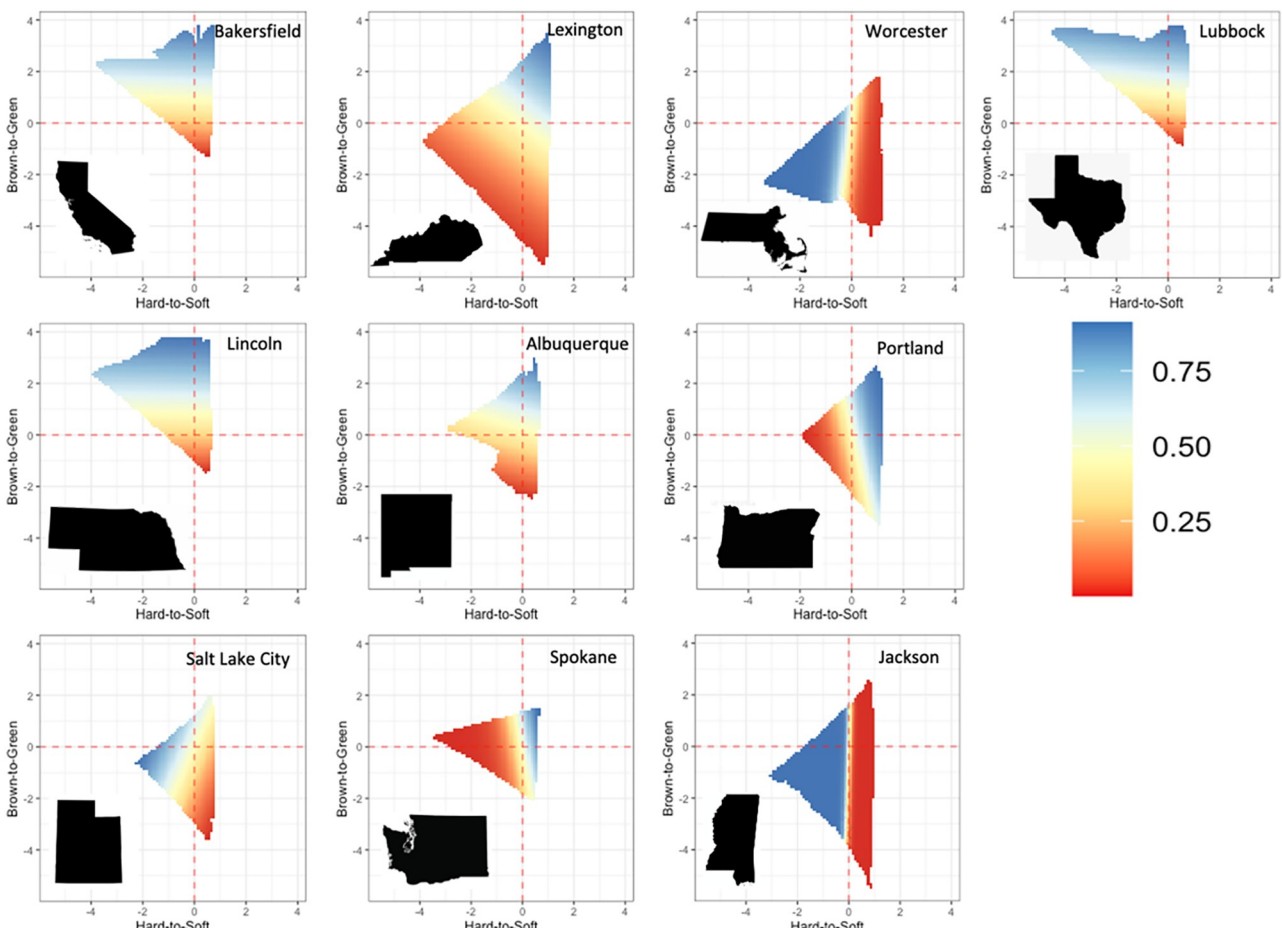

**Fig 4. Predicted robin occupancy along dual landscape gradients: Surface plots depicting robin occupancy on both the brown-to-green (y axis) and hard-to-soft (x axis) for each of the ten cities.** The color scale ranges from low occupancy (red) to high occupancy (blue). Variation in surface plots highlights the differences in landscape composition in study cities, and the variable response to urbanization along the hard-to-soft gradient.

road to train a semi-automated model. Metrics proposed as generalizable for use in human-natural systems also tend to focus on one axis of landscape modification, typically urbanization, rather than the full spectrum of changes to the landscape [64, 65]. Recently, a human modification gradient [66] has been produced that incorporates all aspects of the human footprint, however, it results in a single metric making it difficult to distinguish differential effects of agriculture or urbanization, as we have demonstrated here.

Multiple metrics have been used to analyze and quantify spatial change in human-dominated systems. Large suites of input variables ranging from landscape configuration measures to human population density have been used to identify multiple important features of change in urban landscapes in several notable instances [9, 65, 67, 68]. In most of these cases, however, multiple univariate measures are identified (e.g., using multi-variate analyses) as representative of landscape change along urban-rural gradients. Meanwhile, Berland and Mason [67], noted that dominant factors or principal components could perhaps be used to directly represent urbanization rather than selecting the variables with highest loading. Ultimately, regardless of the number of metrics utilized, or how they were derived, prior multi-metric research has tended to focus on identifying one aspect of landscape change, namely urbanization.

Furthermore, despite identifying multiple important measures, none of these explicitly promote a dual-axis or multi-dimensional application of these measures.

The multi-dimensional landscape gradient approach we propose here offers the flexibility to balance regional adaptability with local specificity and ecological realism to better understand more mechanistically the relationships between landscape structure and ecological processes [58, 69]. We use an established multi-variate statistical approach to succinctly describe spatial heterogeneity and employ readily available NLCD data to incorporate complexity of the entire landscape into a clear and consistent dual axis of human-influence. Although the NLCD dataset is limited to the United States, it employs a nearly identical landscape classification system as other products, including the European Space Agency's GlobCover data [70] or Copernicus Global Land Cover [71], and therefore should be applicable globally wherever such landcover data are available. In addition to PCA, other multi-variate methods have been suggested as alternatives when identifying landscape gradients, such as factor analysis [9], and non-metric multidimensional scaling [72]. More recently, PCA approaches that better account for similarities and differences in multi-group (e.g., multi-city) data have been developed [73], which may be particularly applicable when analyzing landscape gradients in multiple regions simultaneously. Our analysis has demonstrated the importance of considering multiple axes of variation in landscape gradients, and fits within a methodological framework centered on transparent and ecologically informed analysis [3]. It is important to note, however, that city-specific means (i.e., mean effect of landscape on occupancy) in our global analysis may influence interpretation and comparison of effect size between groups. Ultimately, the multi-variate method selected by researchers should be informed by the study's goals, objectives, and types of data available [50]. As the human population continues to grow, the urban, industrial, and agricultural infrastructure must be restructured to ensure future ecological integrity, and the resulting debate over how to effectively do this has led to discussions of land-sharing, i.e., integrating natural systems into the mix of human land-uses, versus land-sparing, i.e., where natural and human systems are concentrated in large, individualized patches. Due to a traditional one-dimensional perspective of landscape heterogeneity, this discussion has largely taken place for agricultural [74], and urban [75] systems in isolation. In reality, however, urban, agricultural, and natural landscapes are inherently inter-mixed. Viewing the land-sharing versus land-sparing debate through a multi-dimensional lens of landscape heterogeneity views the landscape mosaic as a more realistic integrated agro-urban-natural system. Furthermore, the species that will benefit or suffer most from any specific sharing or sparing management, depends entirely on the landscape context within which they are evaluated [76]. Determining how to design a conservation strategy and manage a heterogeneous regional landscape for this species would require that the entire human-natural mosaic be considered and could be facilitated with a multi-dimensional approach to landscape context.

American robins are widely considered to be urban-adaptive and are thought to benefit from urbanized (e.g., *hard*) landscapes with human habitation [77, 78]. However, our results consistently predicted higher occupancy in more forested (*green*) regions over areas predominantly agricultural (*brown*), while the effect of the hard-to-soft axis on robin occupancy varied by city both in terms of magnitude and direction likely due to regional variation in composition of the *soft* landscape (Fig 4). Regional variation in the effect of hard-to-soft on robin occupancy demonstrates the need to consider and decouple multiple dimensions of landscape heterogeneity and suggests that ecological response to human-dominated landscapes is highly nuanced and regionally variable. While highly adaptable and able to exploit many habitat types, robins showed a preference for natural areas in proximity to urbanization (i.e., *green*-and-*hard*) over those in more agricultural landscapes. Our approach synthesizes prior research on the species where single landscape gradients were considered in isolation. In

urban contexts higher presence and survival of robins was reported in residential yards, wood-lots and golf courses [79, 80], while studies in agricultural landscapes found that robins were more common in habitat fragments surrounded by urbanization than those surrounded by agriculture [81].

City-specific variation in robin response to landscape heterogeneity reiterates the importance of landscape context on biotic response. The size (i.e., spatial extent) and density (i.e., human population) of human-dominated landscapes significantly impacts the direction and magnitude of biotic response, with larger and more densely populated cities typically resulting in a stronger negative response [82, 83]. Had our analysis centered on larger or smaller urban regions the specifics of robin response may have differed, however, our core findings—the importance of considering multiple landscape gradients and regional variation in response—would likely have remained. Though, additional research into the impacts of size of human-dominated landscapes on the use of multi-dimensional landscape gradients is warranted. We saw that robin occupancy was demonstrably influenced by both axes of human-modification across the continental United States, suggesting that a continued reliance on one-dimensional landscape descriptors may result in ecosystem pattern being misinterpreted as inherent stochasticity (e.g., noise), when in fact it reflects an overlooked component of the landscape. Specifically in our context, an analysis using a conventional hard-soft gradient would have overlooked the value of green (natural) landscapes integrated in hard (urban) regions for robins (Fig 4). Bearing this in mind, management decisions that consider only a single aspect of the human-natural landscape may overlook or misinterpret ecological response and result in ineffective conservation plans [84].

All measures of landscape heterogeneity are imperfect representations of reality and therefore fall short to varying degrees, and it is unlikely that any single metric will be ideally suited to every question of ecological pattern and process [85]. Therefore, extending one-dimensional descriptors to a multi-dimensional perspective can help move toward a more general understanding of landscape mosaics. And yet, oversimplified one-dimensional measures such as percent forest cover, or percent impervious surface continue to dominate the literature [3]. Multi-city analysis of urban ecosystems has experienced rich growth in recent years. This work has highlighted the negative and positive potential impacts of urbanization on biodiversity, while stressing the importance of the regional landscape context in driving the direction and magnitude of biological response [82, 83, 86]. Still, multi-region analysis remains hampered by inconsistencies in study design and methodological limitations [87, 88]. The multi-dimensional, dual-axis understanding of spatial heterogeneity we describe has the potential to improve and standardize existing approaches to producing ecologically relevant landscape metrics leading to improvements in multi-region research and valuable insight into patterns of ecological response within and across human-dominated systems.

## Supporting information

**S1 File. Supplemental landscape gradient analysis.**
(DOCX)

**S2 File. American robin occupancy analysis: Supplementary tables and figures.**
(DOCX)

## Author Contributions

**Conceptualization:** Benjamin Juan Padilla.

**Formal analysis:** Benjamin Juan Padilla.

**Investigation:** Benjamin Juan Padilla.

**Project administration:** Benjamin Juan Padilla.

**Supervision:** Chris Sutherland.

**Visualization:** Benjamin Juan Padilla.

**Writing – original draft:** Benjamin Juan Padilla.

**Writing – review & editing:** Chris Sutherland.

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
