## [Decision Letter · Decision Letter 0]

17 Jun 2021

PONE-D-21-15482

Defining dual-axis landscape gradients of human influence for studying ecological processes

PLOS ONE

Dear Dr. Sutherland,

Thank you for submitting your manuscript to PLOS ONE. After careful consideration, we feel that it has merit but does not fully meet PLOS ONE’s publication criteria as it currently stands. Therefore, we invite you to submit a revised version of the manuscript that addresses the points raised during the review process.

Although both reviewers had generally favorable perspectives of your manuscript, they both also had a list of concerns.  Most of these are minor and will be straightforward to incorporate.  The major concern identified by both reviewers had to do with the rigor of the PCA analysis.  Please carefully consider whether this analysis is the most appropriate for your data and questions.  Reviewers provided a number of references to consult as you revise the manuscript.

We look forward to receiving your revised manuscript.

Kind regards,

Janice L. Bossart

Academic Editor

PLOS ONE

Journal Requirements:

3. We note that Figure 1 in your submission contain map images which may be copyrighted. All PLOS content is published under the Creative Commons Attribution License (CC BY 4.0), which means that the manuscript, images, and Supporting Information files will be freely available online, and any third party is permitted to access, download, copy, distribute, and use these materials in any way, even commercially, with proper attribution. For these reasons, we cannot publish previously copyrighted maps or satellite images created using proprietary data, such as Google software (Google Maps, Street View, and Earth). For more information, see our copyright guidelines: http://journals.plos.org/plosone/s/licenses-and-copyright.

a) You may seek permission from the original copyright holder of Figure 1 to publish the content specifically under the CC BY 4.0 license.  

Reviewers' comments:

Reviewer's Responses to Questions

**Comments to the Author**

1. Is the manuscript technically sound, and do the data support the conclusions?

Reviewer #1: Yes

Reviewer #2: Partly

2. Has the statistical analysis been performed appropriately and rigorously? 

Reviewer #1: Yes

Reviewer #2: No

3. Have the authors made all data underlying the findings in their manuscript fully available?

Reviewer #1: No

Reviewer #2: Yes

4. Is the manuscript presented in an intelligible fashion and written in standard English?

Reviewer #1: Yes

Reviewer #2: Yes

5. Review Comments to the Author

Reviewer #1: NOTE: The review is written in markdown style format. I've also attached it as a pdf, which may be a little more intelligible.

# Review for:

*Defining dual-axis landscape gradients of human influence for studying ecological processes*

This paper describes a way to separate multiple axes of human modification on the landscape through the use of spatially weighted averaging of NLCD data via a Gaussian kernel and application of PCA, which is simple and provided intuitive results (a win win in my opinion). The statistical analysis as described appears sound, and using a common and easy to identify bird species helps alleviate any concerns I may have potentially had with using community science data from ebird (though you perhaps want to sell other readers on this in the methods section). I pretty much have one primary concern (top-level thought number 1), and then some relatively minor suggestions / improvements to the manuscript. I'm waiving anonymity (I'm Mason Fidino), so if there are any questions that arise from this review please reach out to me at mfidino@lpzoo.org.

## Top-level thoughts

1. One aspect I've wondered about a lot, as someone who is also interested in how species distributions differ within and among cities, is how applicable PCA is when applied to multiple cities combined. Certainly, when applied this way PCA will identify the average gradient of urban intensification across cities, but then we do not know how well that average gradient applies to each city. This can also cause statistical issues in terms of model interpretation. For example, while the global mean of each gradient identified via PCA will be zero (assuming the data has been scaled), the city specific means when subset from that global gradient are not likely going to equal that global mean. Similarly, the standard deviation of each gradient will not be equivalent to the global standard deviation. Depending on the extent of these differences, I would imagine it could be very hard to compare effect sizes of regression coefficients from a global PCA that has then been subset among multiple cities (which is why we generally want to have mean = 0 and sd = 1 for a given continuous covariate).

There are, of course, other styles of PCA that help address this (e.g., multi-group PCA, or more recently network PCA), but it's something I've never really seen used in ecology (but perhaps should be when we start thinking about quantifying gradients across cities).

```

Eslami, A., Qannari, E. M., Kohler, A., & Bougeard, S. (2014). Multivariate analysis of multiblock and multigroup data. Chemometrics and Intelligent Laboratory Systems, 133, 63-69.

Codesido, S., Hanafi, M., Gagnebin, Y., González-Ruiz, V., Rudaz, S., & Boccard, J. (2020). Network principal component analysis: a versatile tool for the investigation of multigroup and multiblock datasets. Bioinformatics.

```

The authors did apply single-city PCAs (available in supplemental material), but the explanation within the main text (and supplemental table) still have not especially assuaged the concerns I have here. For example, the single-city PCAs center on an individual city, whereas the multi-city PCA centers on the global average. How different the average of each variable is among cities is then could possibly have a big influence on what is being observed. At a minimum, perhaps it would help to city-mean center your variables before applying the global PCA (and then not mean centering when calling `princomp()` or `prcomp()` in `R`)?

2. With this many cities, and given the interest in quantifying differences within and among cities, I wonder if it would help to treat city as a random effect within the model. Currently, all regression coefficients have to be interpreted relative to some baseline city for comparison whereas a multi-level model you can estimate an average response and deviation from that response for each city. Of course, this adds in the complexity of needing to do a Bayesian analysis, but even if the authors do not want to specify a model in JAGS, NIMBLE, or Stan the `ubms` package facilitates unmarked style analyses while adding in the ability to include `lme4` style random effects into the linear predictors for occupancy and detection. Definitely not necessary, but something to consider here.

https://kenkellner.com/ubms/index.html

3. Multi-city urban ecology research is still relatively new, and so there are not many papers out yet about how differences among cities may influence what we observe within a city. However, I'd encourage the authors to think about the results they found in the context of other multi-city studies (some that come to mind are below). Please note that while I try my best to not encourage authors to cite my papers in the review process, in Fidino et al. (2020) we specifically quantified how common urban mammals urbanization response differed across cities due to landscape level differences among cities, which I think is right in line what what you are observing with your own analysis.

```

Aronson, M. F., La Sorte, F. A., Nilon, C. H., Katti, M., Goddard, M. A., Lepczyk, C. A., ... & Winter, M. (2014). A global analysis of the impacts of urbanization on bird and plant diversity reveals key anthropogenic drivers. Proceedings of the royal society B: biological sciences, 281(1780), 20133330.

Beninde, J., Feldmeier, S., Werner, M., Peroverde, D., Schulte, U., Hochkirch, A., & Veith, M. (2016). Cityscape genetics: structural vs. functional connectivity of an urban lizard population. Molecular ecology, 25(20), 4984-5000.

Fidino, M., Gallo, T., Lehrer, E. W., Murray, M. H., Kay, C., Sander, H. A., ... & Magle, S. B. (2020). Landscape-scale differences among cities alter common species’ responses to urbanization. Ecological Applications, n/a (n/a), e2253. doi: https://doi. org/10.1002/eap, 2253.

Gagné, S. A., Sherman, P. J., Singh, K. K., & Meentemeyer, R. K. (2016). The effect of human population size on the breeding bird diversity of urban regions. Biodiversity and conservation, 25(4), 653-671.

```

4. How universal these trends observed here are conditional on the cities used. As the paper only focuses on cities that range in populations from 200K to 500K I am still left wondering how applicable this is to smaller or larger cities. Certainly worth a few sentences to explore the idea in the discussion.

## Introduction

---

### Top-level thoughts

1. This manuscript gets right to the point, which is refreshing.

2. There is a little bit of passive voice that creeps in, especially with the use of 'understanding'. A small edit, but it would be nice to see these in the active voice and edit such sentences to remove some unnecessary gerunds.

3. As a near permanent change on the landscape, I often wonder why we equate urban intensity as a "disturbance." (Line 71). Would it be more clear to say that species richness is greatest at intermediate levels of urban intensification (or perhaps human modification to keep with the phrase you are using), when landscape heterogeneity is often at it's greatest?

4. The introduction uses the phrase "ecological processes" enough, which is perhaps a little too vague at times. For example, the manuscript states that fragmentation and human population density are not related to a range of ecological processes (line 75), but it would help to be more specific about what these ecological processes are.

### Line by line comments

Line 41: I understand the point of this sentence, but the wording makes it difficult. I think it is because it is unclear what 'expands' is acting upon, the link between 'human population' and 'ecological footprint', and the assumption that this means natural landscapes are being transformed due to this.

Line 72: replace "are more variable," with "vary"

Line 75: missing a word here.

## Methods:

---

### Top-level thoughts

1. A sentence in the PCA section of the methods wherein you give a heads up that you'll describe these gradients within the results section would be helpful. I got a little confused at what is only presented in the methods, and it had me wondering whether or not the manuscript was actually going to define the primary gradients captured in the analysis, provide the loadings, etc (which was in the results).

2. A little more explanation about what a "location" is could be helpful. Were surveys thinned so you could assume they were spatially independent?

### Line by line comments

Line 119: What do these percentages represent? Composition across all landscapes? If so, that does not indicate that landscape composition is similar. We would instead be interested in how much these dominant landscape types vary among cities. For example, based off of table 1, forest landcover is clearly not similar across cities.

Line 173: At a minimum cite either the MacKenzie et al. book, a paper, etc. for occupancy modeling.

## Results

---

### Top-level thoughts

1. Loadings are just as important as the description of what the gradient represents. I'm assuming that is what is listed in Table 3 (PC1, PC2, and PC3), but a little more explanation in the table header can make that explicit.

2. What are the values presented in parentheses? Regression coefficients and p values? Regression coefficients and standard errors? Please provide a little more information.

3. Figure 4 does not especially capture the variation that is being explained. Something similar to Fig 3 for each city could help with this (perhaps replacing figure 3 if there are space constraints, which is focused on detection with one that is focused on within vs among city variability in occupancy).

## Discussion

---

### Top-level thoughts

Since there was no comparison here to one-dimensional approaches for urban gradients I am not certain how the analysis described here represents a challenge. No doubt there is still merit in the PCA approach used here, but perhaps it would be better to sell the reader on the insights which you may have missed otherwise.

Thinking about it even further, and perhaps I'm getting a little off-topic, but urban ecologists have tried to capture urban intensity in a variety of ways. Traditionally, we used categorical metrics (e.g., urban vs rural) which are not useful for a variety or reasons. Following this we tried more continuous metrics that were simple to calculate like distance to city center, which while better than categories it effectively assumes a city is an onion, steadily decreasing in urbanization from it's core through each "onion layer". Modern cities, however, are not onions, so again not a great approach. Currently, I would argue we are in the "one-dimensional" vs "PCA" style approach like this manuscript details. While I fit into the PCA camp a bit more, I'm still a little agnostic to which approach is better, because both have their uses. For example, I would argue that a one-dimensional approach makes it easier to fit multi-level models because that single gradient can provide you a way to incorporate within vs among city variation as two separate continuous covariates (e.g., average of a given gradient within each city, and then a city-mean centered site-specific gradient, Fidino et al. 2020).

## Figures.

Figure 3 legend says it's about detection but the y-axis says occupancy.

Reviewer #2: I appreciate the efforts in suggesting a standardized approach for defining multi-dimensional gradients, as this is key for understanding species responses to human modifications (as shown for the American Robin) and for informing management and conservation efforts. However, previous attempts in this direction have been made and have already suggested a multi-dimensional approach instead of using one-dimensional measure (e.g. Luck and Wu 2002; Hahs and McDonnell 2006; Schwarz 2010; Modica et al. 2012; Suarez-Rubio and Krenn 2018), thus this should be acknowledged and discuss in the manuscript.

Although most statistical analyses were appropriate, a Factor Analysis (FA) should be performed instead of the PCA. du Toit and Cilliers (2011) carefully argue why PCA is not the correct method when the purpose is identifying uncorrelated axes of variation and distinguishing the different aspects forming a multi-dimensional urbanization gradient ‒despite its earlier use. Additionally, subsequent studies used this methodology in their research e.g. Berland and Manson (2013), Leveau (2013), Suarez-Rubio and Krenn (2018). The manuscript's main conclusions are based on the PCA results, which might change considerably if the data were reanalyzed using FA.

A better description of the methods, in particular the landscape quantification is needed. Although it was mentioned that the landscape analysis followed the framework of a previous publication, it is important for the repeatability of the study to briefly describe how the quantification was performed. In Table 2, a description of decisions is included but it is unclear how the physical land-cover differ from the NLCD and from where the land-use data are coming from. In addition, it seems that the study only included landscape composition variables, why configuration variables are not included, this was shown to be relevant in previous studies (see references in paragraph above).

Minor comments:

Line 28 It would be nice if the name of the focal species is mentioned in the abstract.

Line 102 As “medium size city” have different meanings in other urban settings around the world, it will be relevant to include the average size of the cities studied and in Table 1 include the size of each city.

Line 220 There are several tables included in S1, to which one in particular this refers to?

Line 287 Table 3 or 4?

Line 304-306 Where are these results depicted?

Lines 320-326 I assume that these results are presented in S2 Table 3, if so, please make sure that the numbers are the same as some of them are not (e.g. -1.68 vs. -1.72).

Line 342 I do not think this is supported by Fig. 3.

Line 369 Although the GlobCover data is a good source, it is quite old (2009), so better also to include the Copernicus Global Land Cover.

Lines 351-361 Please refer to the comment above.

Line 383 to which species this refers to, the American Robin?

Table 1: There is no need to include % next to each number under the land-cover classes.

Fig.1 For the international readers not familiar with the location of the cities in the US, including the name of the city in the map or even a number as an ID of the city will be highly appreciated. In addition, it will be better to be consistent in the way you refer to them, sometimes the name of the city is used (Line 78) and sometimes the state where the city is located (Line 318, S2 Table 3).

Fig. 2 In the lower left corner, I think that % of forest should be high and not low.

Fig. 3 Based on the text (Lines 302-304), the y-axis in the figures should be detection probability, correct?

S1 will benefit from including first the information about the spatial extent (city window) and the then the smoothing scale and city specific analysis.

The manuscript is well-written. But there are few things that should be revised:

Line 233 Replace boldened by bold.

Line 236 It should be 11.1%

Line 132 and 400 Instead of including the citation in author-year format include the corresponding number.

References should be checked for accuracy and consistency, e.g. 6, 12, 22, 35, 37, 69.

References:

Berland A, Manson SM (2013) Patterns in Residential Urban Forest Structure Along a Synthetic Urbanization Gradient. Annals of the Association of American Geographers 103:749-763

du Toit MJ, Cilliers SS (2011) Aspects influencing the selection of representative urbanization measures to quantify urban-rural gradients. Landscape Ecol 26:169-181

Hahs AK, McDonnell MJ (2006) Selecting independent measures to quantify Melbourne's urban–rural gradient. Landsc Urban Plan 78:435-448

Leveau LM (2013) Bird traits in urban-rural gradients: how many functional groups are there? J Ornith 154:655-662

Luck M, Wu J (2002) A gradient analysis of urban landscape pattern: a case study from the Phoenix metropolitan region, Arizona, USA. Landscape Ecol 17:327-339

Modica G, Vizzari M, Pollino M, Fichera CR, Zoccali P, Di Fazio S (2012) Spatio-temporal analysis of the urban-rural gradient structure: an application in a Mediterranean mountainous landscape (Serra San Bruno, Italy). Earth Syst Dynam 3:263-279

Schwarz N (2010) Urban form revisited—Selecting indicators for characterising European cities. Landsc Urban Plan 96:29-47

Suarez-Rubio M, Krenn R (2018) Quantitative analysis of urbanization gradients: a comparative case study of two European cities. Journal of Urban Ecology 4:1-14

6. PLOS authors have the option to publish the peer review history of their article (what does this mean?). If published, this will include your full peer review and any attached files.

Reviewer #1: **Yes: **Mason Fidino

Reviewer #2: No

---

## [Decision Letter · Decision Letter 1]

16 Sep 2021

PONE-D-21-15482R1Defining dual-axis landscape gradients of human influence for studying ecological processes

PLOS ONE

Dear Dr. Sutherland,

Thank you for submitting your manuscript to PLOS ONE. After careful consideration, we feel that it has merit but does not fully meet PLOS ONE’s publication criteria as it currently stands. Therefore, we invite you to submit a revised version of the manuscript that addresses the points raised during the review process.

The comments from both reviewers were generally minor so I don’t see any need to send the manuscript out for external review.  That said, both reviewers identified multiple areas where additional detail or modification of existing text will improve readability and/or understanding.  Please explain how you have addressed their concerns in your response. 

Consult the PLOS ONE website and follow the format for cited references.  Carefully check that all references cited within the text are listed in the References Cited section and that all listed references are cited in the text.  Please also pay very careful attention that references are free of any errors.  PLOS ONE articles are not copy edited.

I have listed additional needed changes below:

Tables 1 & 4.  Right justify all numbers.

Table 1.  Remove % signs within body of table.  Add a statement to the caption that numbers are percent cover, or some such.

Table 3.  Why isn’t font size of numbers consistent throughout?  Why are some numbers italicized?

Do a ‘Find’ for all uses of ‘which’.  Ensure that the word preceding which is followed by a comma, e.g. Line 88, a comma should follow ‘approach’. I noticed at least 3 locations were the comma is missing.

Consolidate all information on your focal species and why this particular species into one spot.  Currently, the information is spread out in three spots and is redundant (lines 95-96, 161-162, 188-191).  Also, if you decide you want to refer to the scientific name more than once, then the standard practice is to spell the name of the genus out fully on first use and thereafter abbreviate to just the first letter, i.e. *T. migratorius*.

Line 82. Replace ‘despite the fact that’ with ‘even though’.

Line 157. ‘usinga’ should be ‘using a’.

Line 168-180. Use the same font & size for PCA throughout. My preference is for it to be the same as the general text.

Line 168. Should be ‘PCA is’.

Line 170. Delete the extra space before ‘and’.

Line 171. Add a space after the period.

Line 298. Missing ‘to’.

Line 300. Add a comma after ‘date’.

Line 358. Delete second comma.

Line 366. Should be ‘the human footprint’.

Line 374. Should be ‘the landscape’.

Lines 381-385. Might be better to split into two sentences for clarity.

Line 401. Should be ‘approaches that’.

Line 407. Add a space after the period.

Lines 415-416. ‘Viewing’ and ‘views’?  Maybe reword?

Line 429. ‘suggest’ should be ‘suggests’.

Lines 432-436. Clarity lacking.  Needs modified.  Also, likely better to split into two sentences.

Line 433. ‘species have investigated’??  The species investigated?  I don’t think so.

Line 437. Missing ‘to’.

Line 451. Should be ‘robins’.

We look forward to receiving your revised manuscript.

Kind regards,

Dr. Janice L. Bossart

Academic Editor

PLOS ONE

Journal Requirements:

Reviewers' comments:

Reviewer's Responses to Questions

**Comments to the Author**

1. If the authors have adequately addressed your comments raised in a previous round of review and you feel that this manuscript is now acceptable for publication, you may indicate that here to bypass the “Comments to the Author” section, enter your conflict of interest statement in the “Confidential to Editor” section, and submit your "Accept" recommendation.

Reviewer #1: All comments have been addressed

Reviewer #2: (No Response)

2. Is the manuscript technically sound, and do the data support the conclusions?

Reviewer #1: Yes

Reviewer #2: Yes

3. Has the statistical analysis been performed appropriately and rigorously? 

Reviewer #1: Yes

Reviewer #2: Yes

4. Have the authors made all data underlying the findings in their manuscript fully available?

Reviewer #1: Yes

Reviewer #2: Yes

5. Is the manuscript presented in an intelligible fashion and written in standard English?

Reviewer #1: Yes

Reviewer #2: No

6. Review Comments to the Author

Reviewer #1: Hello,

The revisions the authors made are fantastic. I think the only thing that really needs to be added in the discussion is a bit on how interpretation of city-specific regression coefficients from the global PCA approach. Basically, acknowledge that city-specific means of the PCA may vary from the global mean, and how that can influence interpretation of effect size. This isn't a problem with this approach, but I would hate for someone to get excited by this technique, use it, and then compare effect size among cities via the magnitude of regression coefficients. By giving a reader a roadmap to model interpretation with your approach, you'll hopefully ensure that it gets used effectively and correctly.

Small line by line comments:

Line 179: Provide some justification to the reader about why you also conducted this analysis separately (e.g,. determine how well the combined (i.e., all cities) gradient described city-specific gradients).

Table 1: Shouldn't the numeric data be right justified?

Line 296: You report occupancy probabilities later as values between 0 and 1 instead of a percentage I'd be consistent in the way you report it. My personal preference is [0-1], but that is what it is, a preference.

Line 403: I know what you mean here, but readers unfamiliar with the term 'multi-block' coul be confused. Also, in this case, I think you actually mean 'multi-group'. I believe that multi-block is grouping among covariates (column wise) while multi-group means there is categorical structure of the same covariates (row wise). Maybe modify to "...differences in multi-group (e.g., multi-city) data..."

Figure 3: Remove the underscores in Salt_Lake_City, and perhaps add the state abbreviations (e.g., because there is a Portland, Maine as well).

Cheers,

Mason Fidino

Reviewer #2: Thank you for addressing my previous comments. The clarity of the manuscript has improved and key points were acknowledged and added. However, there are minor things, mostly related to style, that should be considered. This is particularly important as PLOS ONE does not copyedit accepted manuscripts.

L82 Replace ‘despite the fact that’ by ‘although’

L113, 289 Please make sure that the heading included here match the ones included in the Appendix.

Table 1 This was previously mentioned, but the changes were not made. The percentages in each of the columns are not needed, please remove them.

L137-143/Table 2 The description of the methodology included in this paragraph is not clear. I had trouble following the procedure and although you cited Table 2, that did not help. What is the difference between ‘land-cover categories’ mentioned in ‘landscape features’ and the ‘land-cover data’ mentioned for the spatial data? You cited a previous publication, but to allow repeatability of the study it is important to clearly describe these steps.

L286 It should be ‘x’

L354 Remove ‘configuration’. The study does not quantify the configuration of the cities, so including here the word ‘configuration’ is misleading. The information included in Table 1 in S1 Appendix includes only the proportion of land-cover i.e., composition.

L415-417, L437-438 Please revise the grammar of these sentences.

References There are many mistakes in the list of references. I mentioned here only few examples. In some cases each word of the title is capitalized in other instances it is not. In 6, Beggs J. editor, should be removed. In 22 first names should be replaced by initials. #46 is not cited in the text.

7. PLOS authors have the option to publish the peer review history of their article (what does this mean?). If published, this will include your full peer review and any attached files.

Reviewer #1: **Yes: **Mason Fidino

Reviewer #2: No

---

## [Author Response · Author response to Decision Letter 1]

14 Oct 2021

Tables 1 & 4. Right justify all numbers. 

Numbers are right justified

Table 1. Remove % signs within body of table. Add a statement to the caption that numbers are percent cover, or some such.

% signs have been removed, and relevant statement added to caption. 

Table 3. Why isn’t font size of numbers consistent throughout? Why are some numbers italicized? 

Thank you, numbers have been resized and italics removed

Do a ‘Find’ for all uses of ‘which’. Ensure that the word preceding which is followed by a comma, e.g. Line 88, a comma should follow ‘approach’. I noticed at least 3 locations were the comma is missing. 

Thank you for noticing this grammatical error. Corrected

Consolidate all information on your focal species and why this particular species into one spot. Currently, the information is spread out in three spots and is redundant (lines 95-96, 161-162, 188-191). Also, if you decide you want to refer to the scientific name more than once, then the standard practice is to spell the name of the genus out fully on first use and thereafter abbreviate to just the first letter, i.e. T. migratorius. 

We appreciate this suggestion, however, after reviewing the manuscript, we believe that it is best organized as is. The majority of the important life history information on the study species is located in lines 188-191. The information in lines 95 and 96 is a brief reference to the study species in a summary of the manuscript’s objectives, while the reference in lines 161-162 is important information for the statistical methodology described in that section. 

 We have changed the second reference to the genus species name to read as T. migratorious. 

Line 82. Replace ‘despite the fact that’ with ‘even though’. 

Changed

Line 157. ‘usinga’ should be ‘using a’. 

Revised

Line 168-180. Use the same font & size for PCA throughout. My preference is for it to be the same as the general text. 

 All occurrences of PCA have been changed to match the style of the general text.

Line 168. Should be ‘PCA is’. 

Revised

Line 170. Delete the extra space before ‘and’. 

Revised

Line 171. Add a space after the period. 

Revised

Line 298. Missing ‘to’. 

‘To’ has been added, thank you

Line 300. Add a comma after ‘date’. 

Revised

Line 358. Delete second comma. 

Revised

Line 366. Should be ‘the human footprint’. 

Revised

Line 374. Should be ‘the landscape’. 

Revised

Lines 381-385. Might be better to split into two sentences for clarity. 

Revised

Line 401. Should be ‘approaches that’.

Revised

Line 407. Add a space after the period. 

Revised

Lines 415-416. ‘Viewing’ and ‘views’? Maybe reword? 

This text has been reworded for clarity

Line 429. ‘suggest’ should be ‘suggests’. 

Revised

Lines 432-436. Clarity lacking. Needs modified. Also, likely better to split into two sentences. 

The text has been split into two sentences to improve clarity and readability

Line 433. ‘species have investigated’?? The species investigated? I don’t think so. – This has been changed to “research on the species where single landscape gradients were considered in isolation”

Revised

Line 437. Missing ‘to’. 

Revised

Line 451. Should be ‘robins’. 

Revised

The revisions the authors made are fantastic. I think the only thing that really needs to be added in the discussion is a bit on how interpretation of city-specific regression coefficients from the global PCA approach. Basically, acknowledge that city-specific means of the PCA may vary from the global mean, and how that can influence interpretation of effect size. This isn't a problem with this approach, but I would hate for someone to get excited by this technique, use it, and then compare effect size among cities via the magnitude of regression coefficients. By giving a reader a roadmap to model interpretation with your approach, you'll hopefully ensure that it gets used effectively and correctly. 

We have added the following text to lines 410-412 “It is important to note, however, that city-specific means in our global analysis which may influence interpretation and comparison of effect size between groups.”

Small line by line comments:

Line 179: Provide some justification to the reader about why you also conducted this analysis separately (e.g,. determine how well the combined (i.e., all cities) gradient described city-specific gradients). 

Table 1: Shouldn't the numeric data be right justified? � DONE

Line 296: You report occupancy probabilities later as values between 0 and 1 instead of a percentage I'd be consistent in the way you report it. My personal preference is [0-1], but that is what it is, a preference. � we have changed these to 0-1 scale

Line 403: I know what you mean here, but readers unfamiliar with the term 'multi-block' coul be confused. Also, in this case, I think you actually mean 'multi-group'. I believe that multi-block is grouping among covariates (column wise) while multi-group means there is categorical structure of the same covariates (row wise). Maybe modify to "...differences in multi-group (e.g., multi-city) data..." � Thanks for suggestion, this has been added

Figure 3: Remove the underscores in Salt_Lake_City, and perhaps add the state abbreviations (e.g., because there is a Portland, Maine as well). � DONE

Cheers,

Mason Fidino

Reviewer #2: Thank you for addressing my previous comments. The clarity of the manuscript has improved and key points were acknowledged and added. However, there are minor things, mostly related to style, that should be considered. This is particularly important as PLOS ONE does not copyedit accepted manuscripts.

L82 Replace ‘despite the fact that’ by ‘although’ 

Revised

L113, 289 Please make sure that the heading included here match the ones included in the Appendix.

Revised

Table 1 This was previously mentioned, but the changes were not made. The percentages in each of the columns are not needed, please remove them. 

Revised

L137-143/Table 2 The description of the methodology included in this paragraph is not clear. I had trouble following the procedure and although you cited Table 2, that did not help. What is the difference between ‘land-cover categories’ mentioned in ‘landscape features’ and the ‘land-cover data’ mentioned for the spatial data? You cited a previous publication, but to allow repeatability of the study it is important to clearly describe these steps.

We have done our best to improve this portion of the methods, and have revised it as follows “Our decisions regarding the types of landscape features relevant for analysis, the data to represent those features, and the spatial scales of analysis were made to reflect a typical ecological analysis - definitions of, and justification for, these decisions are provided in Table 2” 

L286 It should be ‘x’ 

Revised

L354 Remove ‘configuration’. The study does not quantify the configuration of the cities, so including here the word ‘configuration’ is misleading. The information included in Table 1 in S1 Appendix includes only the proportion of land-cover i.e., composition. 

Revised

L415-417, L437-438 Please revise the grammar of these sentences. 

Revised

References There are many mistakes in the list of references. I mentioned here only few examples. In some cases each word of the title is capitalized in other instances it is not. In 6, Beggs J. editor, should be removed. In 22 first names should be replaced by initials. #46 is not cited in the text. 

Revised

---

## [Editor Report · Decision Letter 2]

22 Oct 2021

PONE-D-21-15482R2Defining dual-axis landscape gradients of human influence for studying ecological processesPLOS ONE

Dear Dr. Sutherland,

Thank you for submitting your revised manuscript to PLOS ONE.  Unfortunately, I'm afraid I need to return it once again given the many minor corrections that are necessary.  As I pointed out earlier, PLOS ONE does not use copy editors.  Please submit a revised version after you've carefully dealt with the needed corrections. I can appreciate that you prefer to leave the information on your case study species in the three separate locations even though it seems redundant to this reader to be told multiple times that your case study species is the American Robin, that the American Robin is *Turdus migratorious*, and that the American Robin is a widespread generalist found in areas with humans.  At the very least, the 2nd mention of the scientific name is unnecessary since the American Robin was already identified as such the first time.  Please delete that second mention.Lines 410-412.  Incomplete sentence:  “It is important to note, however, that city-specific means in our global analysis which may influence interpretation and comparison of effect size between groups.”  Given you're largely, but not entirely, incorporating what the reviewer literally suggested, I'm not exactly sure what changes are needed.  The sentence would easily make sense by simply removing the 'which'.  That said, some readers less familiar with your analyses might appreciate more complete inclusion of the 'city-specific means differing from the global mean' verbiage, as the reviewer included.Many corrections are needed in the references

Use lower case in the titles: L495, L502, L507, L512, L537, L648, L698, L711

Journal name should not be all caps: L529, L526, L547

Italicize scientific names: L571, L680

L562. Delete 1st Ecol

L677. Delete (80-) after 'Science'

L585. Lower case 'Framework'

We look forward to receiving your revised manuscript.

Kind regards,

Janice L. Bossart

Academic Editor

PLOS ONE
---

## [Author Response · Author response to Decision Letter 2]

28 Oct 2021

Dear Janice Bossart, 

Thank you for the detailed editorial review of our manuscript. We have made all final changes, and we trust you will agree that all necessary modifications and edits. Below you will find the final round of revisions, with our edits in red.

1. I can appreciate that you prefer to leave the information on your case study species in the three separate locations even though it seems redundant to this reader to be told multiple times that your case study species is the American Robin, that the American Robin is Turdus migratorious, and that the American Robin is a widespread generalist found in areas with humans. At the very least, the 2nd mention of the scientific name is unnecessary since the American Robin was already identified as such the first time. Please delete that second mention.

Thank you for standing firm on this. We have removed the second reference to the scientific name in line 191. Further, we have removed our description of the species ecology in lines 95 through 96, and it now reads “We demonstrate the utility of our approach in a case study analysis of American robin (Turdus migratorious), occupancy.”

2. Lines 410-412. Incomplete sentence: “It is important to note, however, that city-specific means in our global analysis which may influence interpretation and comparison of effect size between groups.” Given you're largely, but not entirely, incorporating what the reviewer literally suggested, I'm not exactly sure what changes are needed. The sentence would easily make sense by simply removing the 'which'. That said, some readers less familiar with your analyses might appreciate more complete inclusion of the 'city-specific means differing from the global mean' verbiage, as the reviewer included.

We have removed ‘which’ in order to make the sentence complete. Further, to aid in providing the reader with clarity regarding city-specific means, we have added the following to the text: “city-specific means (i.e., mean effect of landscape on occupancy) in our global analysis”

3. Many corrections are needed in the references

Use lower case in the titles: L495, L502, L507, L512, L537, L648, L698, L711

Journal name should not be all caps: L529, L526, L547

Italicize scientific names: L571, L680

L562. Delete 1st Ecol

L677. Delete (80-) after 'Science'

L585. Lower case 'Framework'

All references have been corrected

---

## [Editor Report · Decision Letter 3]

2 Nov 2021

Defining dual-axis landscape gradients of human influence for studying ecological processes

PONE-D-21-15482R3

Dear Dr. Sutherland,

Congratulations!  We’re pleased to inform you that your manuscript has been judged scientifically suitable for publication and will be formally accepted for publication once it meets all outstanding technical requirements.

Within one week, you’ll receive an e-mail detailing any required amendments.  At that time please also correct the one (very) minor remaining issue:  Line 97 - remove the unnecessary comma after "migratorious)," before submitting the final version that will go to press.

When these required issues have been addressed, you’ll receive a formal acceptance letter and your manuscript will be scheduled for publication.

Kind regards,

Dr. Janice L. Bossart

Academic Editor

PLOS ONE
---

## [Editor Report · Acceptance letter]

9 Nov 2021

PONE-D-21-15482R3 

Defining dual-axis landscape gradients of human influence for studying ecological processes 

Dear Dr. Sutherland:

I'm pleased to inform you that your manuscript has been deemed suitable for publication in PLOS ONE. Congratulations! Your manuscript is now with our production department. 

Kind regards, 

on behalf of

Dr. Janice L. Bossart 

Academic Editor

PLOS ONE